# Use of hybrid quantum-classical algorithms for enhancing biomarker classification

Aninda Astuti[1‡], Pin-Keng Shih[2,3‡], Shan-Chih Lee[4‡], Venugopala Reddy Mekala[5‡], Ezra B. Wijaya[1,6,7‡], Ka-Lok Ng[1,8,9*]

1 Department of Bioinformatics and Medical Engineering, Asia University, Taichung, Taiwan, 2 School of Medicine, China Medical University, Taichung, Taiwan, 3 Department of Surgery, China Medical University Hospital, Taichung, Taiwan, 4 Department of Medical Imaging and Radiological Sciences, Chung Shan Medical University, Taichung, Taiwan, 5 Dan L Duncan Institute for Clinical and Translational Research, Baylor College of Medicine, Houston, Texas, United States of America, 6 Reproductive Health Center, Hualien Tzu Chi Hospital, Buddhist Tzu Chi Medical Foundation, Hualien, Taiwan, 7 Department of Physiotherapy, Faculty of Health Science and Technology, Binawan University, Jakarta, Indonesia, 8 Quantum Computing Research and Development Center, Asia University, Taichung, Taiwan, 9 Department of Medical Research, China Medical University Hospital, China Medical University, Taichung, Taiwan

‡ These authors share co-first author on this work. VRM and EBW share co-second author on this work.
* klng@asia.edu.tw, ppiddi@gmail.com

## Abstract

Quantum machine learning (QML) combines quantum computing with machine learning, offering potential for solving intricate problems. Our research delves into QML's application in identifying gene expression biomarkers for clear cell renal cell carcinoma (ccRCC) metastasis. ccRCC, the primary renal cancer subtype, poses significant challenges due to its high lethality and complex metastasis process. Despite extensive research, understanding the mechanisms of cancer cell dissemination and establishment in distant sites remains elusive. Identifying metastasis biomarkers is a daunting task in machine learning. Our study addresses the need for improved execution time and accuracy in QSVC and QNN algorithms compared to SVC and NN for binary classification. Drawing inspiration from the Neural Quantum Embedding (NQE) method, we propose a two-stage approach for the binary classification problem. We aim to assess if integrating NQE with QSVC/QNN enhances performance compared to NQE with SVC/NN across diverse biomedical datasets, demonstrating the effectiveness and generalizability of the approach.

## 1. Introduction – quantum machine learning

Machine learning (ML) is a sub-area of artificial intelligence, enabling computers to learn from data and improve tasks over time. ML has a very wide range of applications, including quantum many-body problems [1], phase transition [2], classification and regression [3], engineering [4], bioinformatics analysis [5], anomaly detection [6], bibliometric analysis of quantum k-means clustering [7] etc.

**Data availability statement:** The raw datasets and codes for the NQE+QSVC and NQE+QNN algorithms are available under https://github.com/anindaastuti/NQE-QSVC-QNN. It is designed primarily for academic purposes rather than commercialization.

**Funding:** Miss Aninda Astuti and Dr. Ka-Lok Ng works are supported by (i) National Science and Technology Council (NSTC) (grant numbers: NSTC 112-2221-E-468-021), and (ii) Asia University and China Medical University Hospital (grant number: ASIA-112-CMUH-6). The funding bodies were not involved in the design of the study, in the collection, analysis of the data, or in writing of the manuscript.

**Competing interests:** The authors have declared that no competing interests exist.

Quantum machine learning (QML) [8] is an emerging and popular research field that integrates the concepts of quantum computing and machine learning. This integration leverages the unique capabilities of quantum computers to enhance and accelerate machine learning tasks. The most commonly used model in quantum computing is the gate-based quantum computation model [9]. Essentially the model can be described in three steps: (i) Initialization: Qubits are initialized in a specific quantum state, often the |0> state. (ii) Encoding: Input data is encoded into the quantum amplitudes of these qubits. (iii) Quantum Gates: Quantum unitary gates are applied to transform the quantum states of the qubits. These gates can operate on multiple qubits simultaneously, a property known as quantum parallelism.

By designing and implementing quantum circuits (quantum algorithms) appropriately, the quantum states undergo interference. This interference enhances the amplitudes of the desired solutions. The final solution is obtained by performing measurements on the quantum states. Due to the inherent randomness in quantum measurements, multiple repetitions are necessary to determine the required solution with high probability.

Quantum computing excels in solving specific types of problems, such as quantum simulations in physics and chemistry. However, it is not expected to replace classical computing entirely. Instead, a hybrid approach that combines classical and quantum computing is often used to tackle complex problems more efficiently [10–12].

Leveraging quantum computing in machine learning may be essential for solving complex problems for chemistry and physics [13], in molecular biology, such as: cancer classification [14–17], non-small cell lung cancer classification [18], COVID-19 [19], biochemical systems [20], medical imaging [21,22], malignant breast cancer diagnosis [23], pancreatic cancer data classification [24], cancer gene expression biomarker [25], oncology application [26], clinical decision [27], drug discovery [28–30] etc.

Renowned ML algorithms, including the support vector machine classifier (SVC), random forest, neural network (NN), ensemble methods etc. Among the many well-known ML algorithms, SVC can achieve a better performance in certain real-world applications [31–34].

A quantum version of the SVC algorithm (QSVC) was proposed by Rebentrost et al., at 2014 [35]. Later, the kernel-based classifier was implemented on a superconducting quantum machine [36]. At 2020, Suzuki et al., [37] suggested to combine different kernels to construct a more efficient feature map for classification using QSVC. QSVC stand out as innovative method that combine the power of quantum mechanics with domain of machine learning [38].

In addition to QSVC, quantum versions of several machine learning algorithms have been developed, including QNN (quantum neural network) [39] and QCNN (quantum convolutional neural network) [40]. Both QSVC and QNN classifiers are considered in the work.

The use of QCNN combined with NQE [41] for classifying the digits 0 and 1 within the MNIST dataset has been reported. It was found that the classification accuracy of QCNN trained with NQE significantly improves compared to QCNN without

NQE training. However, there are no studies on the use of NQE+QSVC and NQE+QNN for classification problems. We explore this possibility using biomedical data. We noted that the quantum version of random forest (QRF) is currently not available.

For both QSVC and QNN, the key factor that determines the success of classification performance is the choice of an appropriate feature map. If the feature map is not well-suited to the data, it will hinder the performance of the entire machine learning model [42].

## 1.1. Classification tasks

In this study we explore quantum algorithms in gene expression biomarker study for clear cell renal cell carcinoma (ccRCC) metastasis. Over several decades, ccRCC is the predominant subtype of renal cancer, accounting for 80–90% of renal cases and remaining a highly lethal urological malignancy [43–45]. Metastasis, the final stage of cancer, remains a challenging area of study despite recent research efforts [46]. The precise mechanisms underlying the departure of cancer cells from their primary site, their dissemination through the bloodstream [47], and their establishment in distant secondary sites are still largely unknown [48]. The identification of biomarkers in metastasis cancer research remains challenging in machine learning research [49,50]. Fares et al. [51] and Hanahan and Weinberg [52] have contributed valuable insights to our understanding of metastasis through their comprehensive reviews.

In our prior study, we observed that the QSVC algorithm has limitations in execution time. It also showed lower classification accuracy compared to SVC for supervised binary classification tasks [53]. We took inspiration from earlier studies on quantum graph neural networks [54], hybrid classical-quantum neural networks [55], and the Neural Quantum Embedding (NQE) method [41]. Based on these, we designed a two-stage process to improve classification performance.

Initially, we employ NQE for data encoding, followed by the utilization of kernel-based algorithms to tackle binary classification problems. Our study will evaluate the performance of the NQE framework combined with QSVC (NQE+QSVC). It will compare this to the performance of NQE combined with SVC (NQE + SVC) on three biomedical datasets. In below, we demonstrate the effectiveness of the process combining NQE with QSVC.

To evaluate the impact of NQE on classification performance other than SVC, we have included studies on NQE + NN and NQE + QNN in this work, utilizing 5-fold cross-validation. For the NN classifier, the following three hyperparameters were optimized: the number of hidden layers, learning rate, and epochs. For the QNN classifier (SampleQNN), the following three hyperparameters were optimized: the number epochs, the number of ZZfeatureMap repetitions, and the number of ansatz layers. The NQE + NN and NQE + QNN models were then applied to the test set.

## 1.2. Noisy intermediate-scale quantum

The persistent challenge of qubit decoherence due to environmental noise remains a significant obstacle in quantum computing. There have been advancements in Quantum Error Correction (QEC) [56], Quantum Error Mitigation (QEM) [57,58], and dynamical coupling techniques [59]. However, building a fault-tolerant quantum computer still appears to be a distant goal.

Noisy intermediate-scale quantum (NISQ) devices [60,61] offer a near-term solution to current challenges. They may show quantum advantage for some tasks, but they still experience errors. Hybrid classical-quantum algorithms are made to run on NISQ computers. They use the strengths of both classical and quantum computing [62] to help overcome qubit decoherence limitations.

## 2. Materials and methods

In this study, we improve the methodology used in our prior research [53], aiming to overcome specific shortcomings and enhance both the robustness and precision of the QML approach. Previous analyses revealed that the QSVC algorithm

had a prolonged execution time and suffered from lower classification accuracy. Enhanced classification accuracy for metastatic patients can improve treatment efficacy and outcomes, and streamline resource allocation in healthcare.

The workflow diagram for the gene expression biomarker study is depicted in Fig 1.

The miRNA classification study does not require PCA. Data imbalance handling is also unnecessary because the smaller number of features and both the disease and control groups have similar sample sizes.

## 2.1. Input data, data balancing and dimension reduction

In order to demonstrate the effectiveness of the present approach, three distinct types of transcriptomic datasets are utilized: (i) gene expression profiles for primary tumor samples (M0) and metastasis (M1) samples, (ii) microRNA (miRNA) expression profiles for normal (NT) and primary (M0) samples with 4 features and (iii) miRNA expression profiles for primary cancer (TP without M1, that is, M0) and metastasis (M1) samples with 14 features, are included in the analysis.

For the gene expression profiles of patients with ccRCC, a total of 424 and 78 primary (non-metastasis, M0) and metastasis (M1) samples were obtained from the UCSC Xena database (http://xena.ucsc.edu/). Due to the M0 samples being 5.44 times larger than the M1 samples, there exists a potential classification bias.

Without applying SMOTE to address the data imbalance, classification performance using 10, 11, and 12 principal components (PCs) is problematic. A key observation is that specificity is perfect (1.0) for both 10 and 12 PCs, meaning

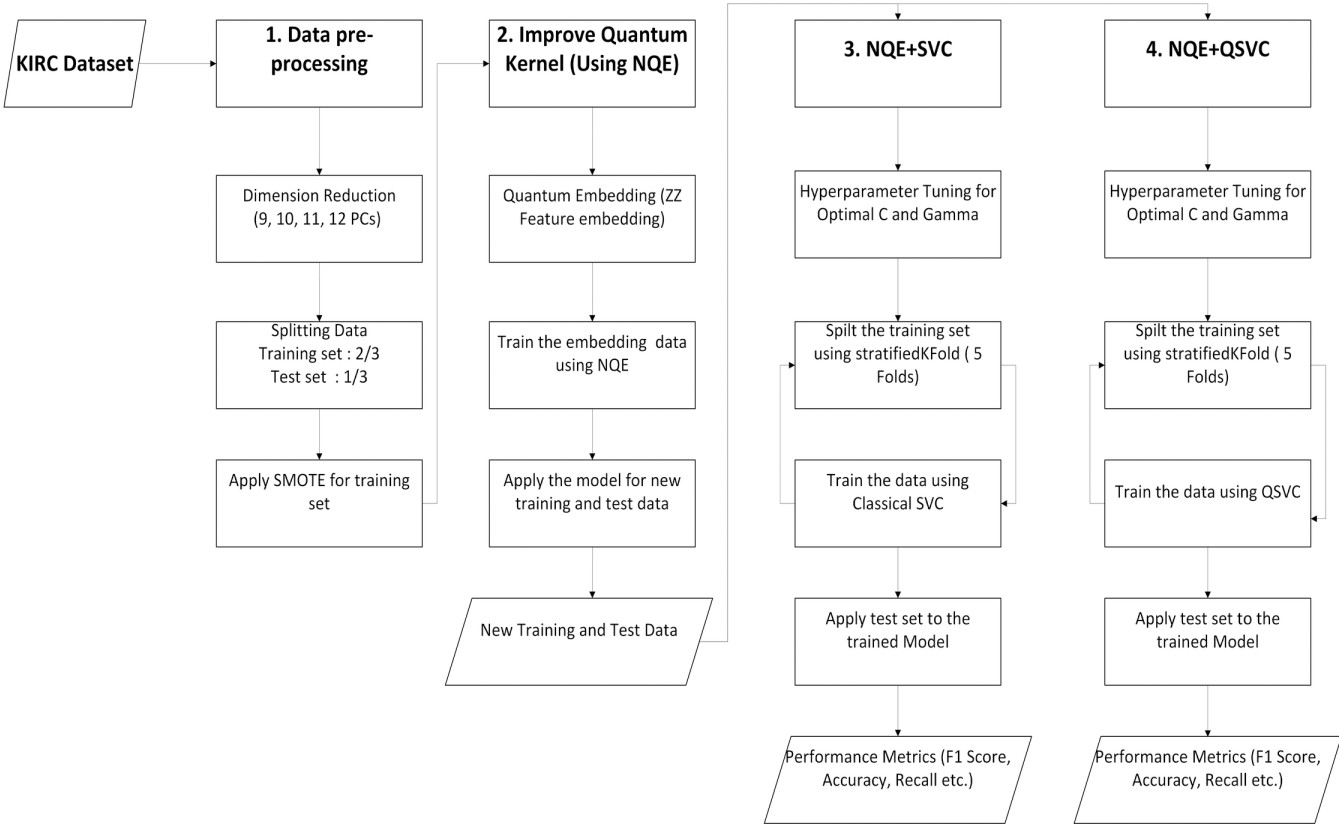

**Fig 1. The workflow diagram for the gene expression biomarker study which involve using principal component analysis (PCA) to reduce the number of features (principal components), thus requiring fewer qubits.** We utilize the SMOTE package for data balancing and optimize SVC and QSVC models through the NQE approach. This hybrid method combines quantum data encoding with neural network training to separate the classes of data into orthogonal subspaces, followed by performance comparison.

the classifier correctly identifies all true negatives (TN, non-metastatic cases, M0). However, recall is very low, dropping as low as 0.03, indicating that the classifier is only able to identify a small fraction of metastatic cases (M1). Given these results, we decided to apply the SMOTE technique to address the data imbalance issue in our current work.

We also have studied the results of using NQE without addressing data imbalance (mRNA expression data using 4 PCs). The results are unstable; for example, there is a significant variation in the F1 Score, ranging from 0 to 0.062, and the MCC can even be negative.

Data balancing algorithms were applied to create a more even distribution between the M0 and M1 classes. This helps machine learning models perform more effectively on the datasets. The oversampling technique is employed to address data imbalance problem by using SMOTE (Synthetic Minority Over-sampling Technique) algorithm [63].

From our prior work [64], we identified 48 differentially expressed genes. This was done using the DEseq2 package [65],with a |log$_2$ Fold Change| (logFC) ≥ 2 and an adjusted p-value below 0.05 (S1 File). By utilizing the principal component analysis (PCA) method, the required qubits decreases from 48 to just 4–12. The covariance matrix has a total of 48 eigenvalues. The top four, five, six, nine, and 12 eigenvalues contribute 52.4%, 56.4%, 59.2%, 69.1%, and 74.8% of the total, respectively. This means that more than 50% of the variance is explained by these eigenvalues. The values indicate the proportion of total variance explained by the first 4–12 principal components. Higher values indicate that the corresponding principal components capture more of the variance in the data, while lower values suggest less contribution to the overall variance.

For the miRNA expression profiles, we examined the expression of mature miRNA strands in ccRCC samples sourced from the TCGA-KIRC project. RNAseq-IlluminaHiseq data was extracted via UCSC Xena. We acquired Level 3 NGS data for dataset (ii), consisting of 72 normal (NT) samples and 170 primary (TP without M1) samples, and for dataset (iii), comprising 170 primary and 40 metastasis (M1) samples. The R 'limma' package was employed to compare miRNA expression between metastasis and primary samples. Significance was determined using the FDR adjustment method, where a p-value < 0.05 indicated statistical significance. As the ratio between primary and M1 samples is less than five, we refrain from employing a data imbalance algorithm.

## 2.2. Data embedding by using Neural Quantum Embedding (NQE)

The goal of data embedding is to train the input features in order to maximize data separability. Improving data separability in higher dimensions enhances training performance. Quantum embedding uses unitary operators to encode classical data non-linearly into high-dimensional quantum state spaces [66,67]. However, achieving this through quantum embedding necessitates introduce more noise and resources, leading to increased circuit depth and gates.

In quantum machine learning, quantum embedding is important because it maps classical data into high-dimensional quantum states, enabling the exploitation of quantum advantages in pattern recognition and classification. The Neural Quantum Embedding (NQE) approach [41] utilizing a hybrid classical quantum approach, that is, combining neural network (NN) and a quantum circuit, to train the input data enables the generation of the feature maps capable of separating the classes of data into orthogonal subspaces. This process is crucial as it enhances classification performance, and the use of a hybrid architecture can also help reduce computation time.

In layman terms, the input features are learned by a NN with trainable weights and a loss function (mean squared error) designed to maximize the separation between the two classes. The outputs are non-linearly mapped to a higher-dimensional Hilbert space using a quantum feature mapping function, which consists of an exponential operator made up of non-linear functions and the rotational Pauli Z operators.

The initial two classes of input samples are encoded into quantum states, and the resulting states are measured to determine whether the two samples share the same class label. This overlap is quantified by the fidelity function. The fidelity result is then used to update the NN's weighting factors, optimizing the separability of the two classes of data into orthogonal subspaces, thereby maximizing their difference. Once iterations complete, save the optimized model, apply it to training and test sets, then pass to the classifier.

Technically, the NQE method involve the following steps. (I) Input feature $x$ is trained by a NN, the output function of the NN is denoted by $g(\mathbf{x}, \mathbf{w})$, where $\mathbf{w}$ represents $r$ trainable parameters. (II) The output function $(g: R^{in} \times R^r \to R^{out}, \ out \ < \ in)$ is inserted into a unitary quantum circuit, which is denoted by $V(g(x, w))$. The unitary operator $V(\phi(x))$ is defined by

$$V(\varphi(\mathbf{x})) = \exp\left( i\sum_k \varphi_k(x)Z_k + i\sum_{j,k} \varphi_{j,k}(x)Z_jZ_k \right) \tag{1}$$

where $\varphi_k$, $\varphi_{jk}$ and $Z$ denote the mapping function for feature $\mathbf{x_k}$, the mapping function for features $\mathbf{x_j}$ and $\mathbf{x_k}$, and Pauli $Z$ operator, respectively.

In general, there are no limits on the type of quantum embedding circuit that can be used. In this case, the operator focuses on improving the ZZ feature embedding by training an Instantaneous Quantum Polynomial (IQP) function, $V(\phi(x))$ [68] through multiple steps. IQP is a model of quantum computing where all the gates in the circuit can be applied in any order without changing the outcome of the computation.

(III) Given the embedding, the initial input feature is encoded to a quantum state, that is, $x \to |x> = V(g(x, w))|0>^{\otimes n}$. The resulting quantum state $|x>$ is measured, the post-process data is used to update the trainable parameter $\mathbf{w}$, by computing a loss function, which is derived from a fidelity measure function $F$,

$$F((\mathbf{x_j}, y_j), (\mathbf{x_k}, y_k)) = [\, |\langle \mathbf{x_j}|\mathbf{x_k}\rangle|^2 - 0.5(1 + y_jy_k)]^2 \tag{2}$$

where $y_k$ denotes the class label of the data $\mathbf{x_k}$. The fidelity loss function can be computed using the swap test [69]. It is noted that if $\mathbf{x_j}$ and $\mathbf{x_k}$ are similar and share the same class label, the loss function is minimized ($\langle \mathbf{x_j}|\mathbf{x_k}\rangle \sim 1$ and $y_jy_k = 1$, so $F \sim 0$) Conversely, if they belong to different classes, the loss function is maximized. Therefore, the loss function offers a means to infer the separability of the data. (IV) The result from (III) is used as input for the NN. Then, the whole process is re-iterate again, and the quantum state is updated to $V(g(\mathbf{x}, \mathbf{w})) |\psi(x)>^{\otimes n}$. (V) In this step, the algorithm returns back to step (I) to update $g(\mathbf{x}, \mathbf{w})$. Fig 2 illustrates each step of NQE in detail. The NQE algorithm is available as an open-source program, accessible for download from https://takh04.github.io/NQE_tutorial.html.

The authors of reference [41] verify NQE's efficacy by experimenting on IBM quantum devices with the MNIST data, elevating classification accuracy significantly from 0.52 to 0.96.

## 2.3. NQE with QML and hyperparameter optimization

### 2.3.1. NQE with SVC, QSVC and hyperparameter optimization. After the two classes of data are successfully separated into different subspaces, we can test how well different machine learning classifiers perform. Fig 3 shows how NQE is combined with QML classifiers, QSVC and QNN, for classifying biomedical data. Fig 3 also illustrates the workflow of integrating NQE with QSVC and QNN classification.

The key design choices are as follows: (i) PCA is used to reduce the number of qubits because simulating quantum circuits on a classical computer requires a lot of computing power. (ii) NQE separates the two classes of data into different subspaces based on a loss function. (iii) Quantum versions of SVC, NN, k-nearest neighbors, and CNN are available, but quantum random forest and other classifiers are not yet supported at the moment. (iv) QSVC and QNN can be implemented using IBM Qiskit. The ZZ feature map and combinations of layers, including rotation and CNOT gates, are used for the QSVC and QNN classifiers, respectively. These quantum classifiers excel in representing complex functions due to their superior expressive power, surpassing the capabilities of classical classifiers [70].

SVC is a powerful algorithm used for classification tasks. It maps features from a lower dimension space to a higher dimension space using a kernel function, subject to a constraint that maximizes the margin between two classes [71,72].

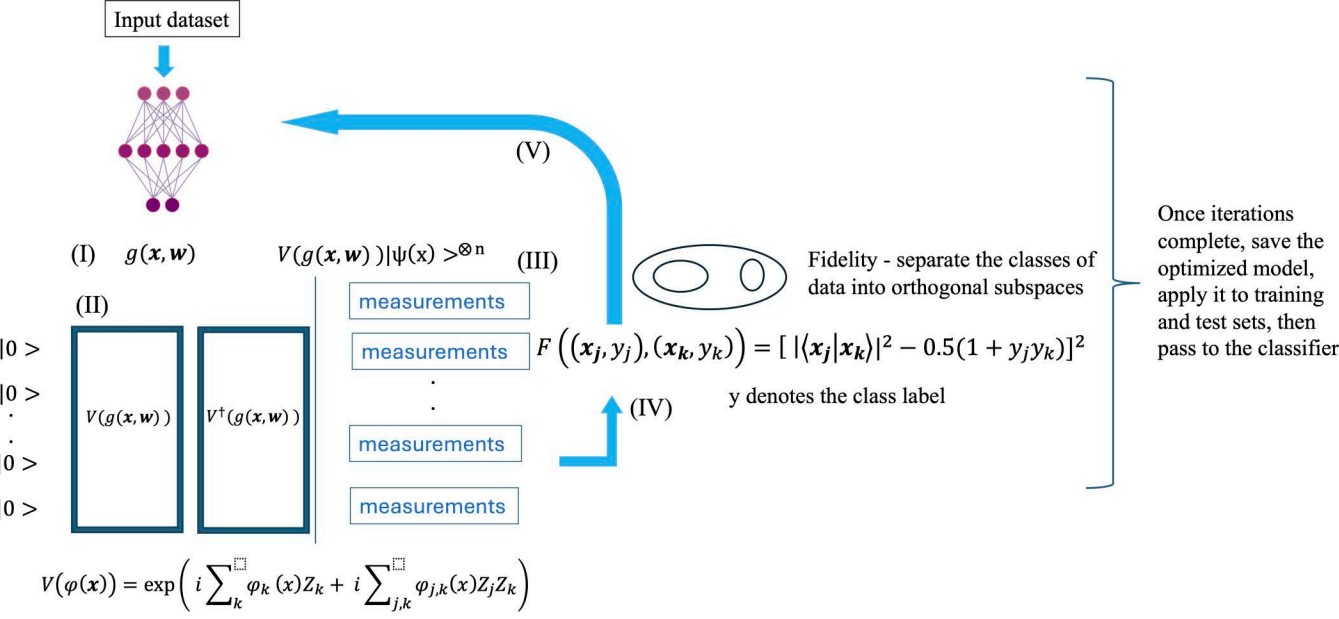

**Fig 2. Data embedding by using Neural Quantum Embedding (NQE).**

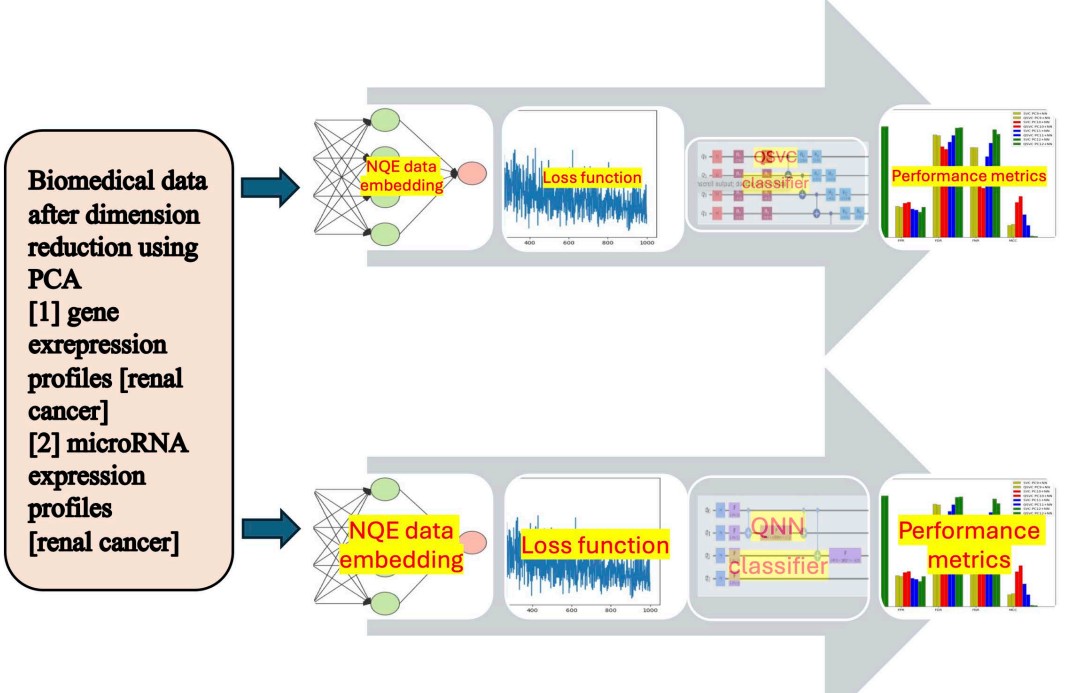

**Fig 3. The workflow of integrating NQE with QSVC and QNN classifiers.**

The margin is the distance between the decision boundary and the closest data point. Mathematically, maximizing the margin is equivalent to minimizing the following function,

$$\min_{w,\,b} \frac{||w||^2}{2}$$

(3)

subject to the constraint

$$y_k\,(w*x_k-t) \geq 1, \quad 1 \leq k \leq n$$

(4)

where $y_k$, $|w|$ and $t$ denote the class label for the *kth* feature, 1-norm of the normal vector to the decision boundary (hyperplane) and the decision threshold (bias), respectively. The above quadratic constraint optimization problem, also known as the primal problem, can be solved by using the method of Lagrange multipliers.

The primal problem is a convex problem; it can be solved efficiently and has the property that any local minimum is also a global minimum. It is possible to derive the dual problem from the primal problem by eliminating $w$ and $t$, then formulating the Lagrange function entirely in terms of the Lagrange multipliers. The dual problem is to maximize the following Lagrange function

$$\sum_{k=1}^{n} \alpha_k - \frac{1}{2} \sum_{j,k=1}^{n} \alpha_j \alpha_k y_j y_k x_j \cdot x_k$$

(5)

subject to the positivity constraints and an equality constraint:

$$\alpha_k \geq 0 \ and \ \sum_{k=1}^{n} \alpha_k y_k = 0$$

(6)

where $\alpha_k$ denotes the Lagrange multipliers. The dual formulation leads to the identification of support vectors. Support vectors are data points that are closest to the decision boundary and are critical for determining the decision boundary. Under the conditions known as the Karush-Kuhn-Tucker conditions, the solutions to the primal and dual problems become equal [73].

The dual formulation is useful because it expresses the Lagrange function using the dot product or kernel function between feature vector pairs. This allows flexibility to choose different kernels, like linear, polynomial, or Gaussian RBF, based on the data and problem. This flexibility enables SVC algorithms to capture complex patterns in data that might not be linearly separable in the original feature space. Solving the dual problem has a time complexity of $O(N^3)$ [74].

The QSVC in the IBM Qiskit package is used to handle the classification problem by leveraging the computational advantages of quantum computers. The QSVC inherits methods like '*fit*' and '*predict*' from Scikit-learn. The QSVC uses a quantum kernel, named '*FidelityQuantumKernel*', for computing the kernel matrix, which can capture complex patterns in the data by leveraging quantum computation.

In QSVC, the quantum kernel $K(x, z)$ is found by calculating the inner product of the quantum states that encode the data, which is given by

$$K(x,z) = \left| {}^{\otimes n}<0|U^{\dagger}(z)U(x)\,|0>^{\otimes n} \right|^2$$

(7)

where $U(x)$ represents the unitary operation derived from the application of a layer of Hadamard gates followed by the quantum feature map using NQE method. The kernel is computed by selecting $U(x)$ and iterating through the other state $U^{\dagger}(z)$. When $U(x)$ is close to $U(z)$, the circuit exhibits a large transition amplitude between states $U^{\dagger}(z)U(x)\,|0>^{\otimes n}$ and ${}^{\otimes n}<0|$, where $|0>$ is the computational z-basis. When the measurement is repeated, the frequency of the all $|0>^{\otimes n}$

state approximates the quantum kernel $K(x, z)$. Once this kernel is trained, it can be used to compute the kernel function between the training and test set, facilitating classification.

We note that the potential advantage of using quantum kernels lies in their ability to leverage computational features that are difficult to simulate efficiently on classical systems. However, this advantage alone is a necessary condition but it is not sufficient [75].

Note that we perform the quantum computing part by using a classical computer to simulate the quantum circuit, rather than executing the code on actual quantum computing hardware.

In order to ensure a fair comparison between the SVC and QSVC classification results, we employ grid search twice to identify the optimal gamma and $C$ parameters in both the SVC and QSVC algorithms. We start with a grid search over a wide range of gamma and C values to get initial estimates. Then, we perform a second grid search to refine these results and find the optimal gamma and C values. The optimal result is determined using the F1 score with five-fold cross-validation.

**2.3.2. NQE with QNN and hyperparameter optimization.** To evaluate the impact of NQE on classification performance other than SVC, we have included studies on NQE + NN and NQE + QNN in this work, utilizing 5-fold cross-validation. The workflow diagram of this study is similar to that in Fig 1, with SVC/QSVC replaced by NN/QNN; therefore, the diagram is omitted. For the NN classifier, the following three hyperparameters were optimized: the number of hidden layers, learning rate, and epochs. For the QNN classifier (SampleQNN), the following three hyperparameters were optimized: the number epochs, the number of ZZfeatureMap repetitions, and the number of ansatz layers. The NQE + NN and NQE + QNN models were then applied to the test set.

## 2.4. Performance metrics

Lastly, the performance metrics results entail the comparison of SVC classifier and the LS-QSVC. To assess the performance of NQE+SVC and NQE+QSVC, we will compute the following 10 performance metrics. It's important to highlight that lower values of the FPR, FDR, and FNR indicate better performance.

$$\text{Sensitivity/Recall}: \ \text{TPR} \ = \ \text{TP} \ / \ (\text{TP} \ + \ \text{FN}) \tag{8}$$

$$\text{Specificity}: \text{SPC} = \ \text{TN} \ / \ (\text{FP} \ + \ \text{TN}) \tag{9}$$

$$\text{Precision}: \text{PPV} = \ \text{TP} \ / \ (\text{TP} \ + \ \text{FP}) \tag{10}$$

$$\text{Negative Predictive Value}: \text{NPV} = \ \text{TN} \ / \ (\text{TN} \ + \ \text{FN}) \tag{11}$$

$$\text{False Positive Rate}: \text{FPR} = \ \text{FP} \ / \ (\text{FP} \ + \ \text{TN}) \tag{12}$$

$$\text{False Discovery Rate}: \text{FDR} = \ \text{FP} \ / \ (\text{FP} \ + \ \text{TP}) \tag{13}$$

$$\text{False Negative Rate}: \text{FNR} = \ \text{FN} \ / \ (\text{FN} \ + \ \text{TP}) \tag{14}$$

$$\text{Accuracy}: \text{ACC} \ = \ (\text{TP} \ + \ \text{TN}) \ / \ (\text{TN} + \text{FP} + \text{FN} + \text{TP}) \tag{15}$$

$$\text{F1 Score}: \text{F1} \ = \ 2 * \text{TP} \ / \ (2 * \text{TP} \ + \ \text{FP} \ + \ \text{FN}) \tag{16}$$

Matthews Correlation Coefficient (MCC):

$$\text{MCC} = \text{TP} * \text{TN} - \text{FP} * \text{FN} / \ \text{sqrt}((\text{TP} + \text{FP}) * (\text{TP} + \text{FN}) * (\text{TN} + \text{FP}) * (\text{TN} + \text{FN})) \tag{17}$$

## 2.5. Data analysis

We use the *Jupyter* Notebook and *PennyLane* (https://pennylane.ai/, a cross-platform Python framework for programming of quantum computing) for developing all the codes and executing them on IBM quantum simulators. To compare the performance of SVC and QSVC, two Scikit-Learn algorithms, PCA and SVC, were implemented on a conventional computer for accomplishing the classification task.

## 3. Results

After completing the PCA for the gene expression profile study, the original $\log_2$(Fold change), that is, $\log_2$FC, values were projected onto the new coordinates (PCs). Assuming we keep nine PCs, the data distribution projected onto two principal components (S1 Fig). The high degree of overlap between the two classes in the PCA plots suggests that the PCs do not provide enough discriminatory power to separate the classes. The overlap between the two labelled classes is still persists even when considering 10–12 principal components (PCs), as illustrated in S2–S4 Figs. This suggests the original features do not capture the underlying differences between the classes effectively; hence, that is a *non-trivial* classification problem. Nevertheless, we will compare the classical and quantum versions to determine which performs better.

In the next step, we divided 2/3 of total sample as training set: 334 samples (286 M0 and 48 M1) and 1/3 as testing set: 168 samples (138 M0 and 30 M1). After employing SMOTE for oversampling the ccRCC gene expression training dataset, we obtain 286 samples for both M0 and M1 as the training set. Next, we perform hyperparameter tuning by dividing the oversampling set into the training set and the validation set using 5-fold cross-validation.

For the miRNA expression profiles, only four features are selected by using multi-variable Cox analysis in our previous study [76]. In the present case, there is no need to use the PCA method for dimension reduction. The data distribution projected onto the four features is shown in S5 Fig. Unlike the mRNA case study, the extent of overlap between the classes is relatively minor; therefore, we anticipate that the classification performance would be better. Next, we divided 2/3 of total sample as training set: 113 primary samples and 27 M1 samples, and 1/3 as testing set: 70 samples.

We utilized the NQE algorithm to pre-process the three input transcriptomic datasets, aiming to optimize data embedding by training the NN part of the NQE algorithm. The number of hidden layers, nodes per layer, the learning rate, the number of iterations and batch sizes are set equal to 1, two times the number of features (=8 nodes), 0.01, 1000 and 10, respectively. The number of qubits corresponds to the number of the input features.

Fig 4 illustrates the quantum embedding circuit. Its objective is to train the input features in a way that maximizes the separability between the M0 and M1 class labels. The number of qubits corresponds to the number of input features (or PCs). The embedding process employs Hadamard (H) gates, rotation (RZ) gates, and control-NOT (CNOT) gates, applied three times. These gates serve the following purposes: (i) creating superposition in the input quantum qubits, (ii) parametrizing the states by applying phase shifts, which are essential for generating interference among the quantum states, and (iii) introducing entanglement between the qubits.

To reverse the embedding, a layer of the inverse of the RZ gates is applied in reverse order. The two mapping functions, $\varphi_k$ and $\varphi_{jk}$ (as shown in the function $V(g(\boldsymbol{x}, \boldsymbol{w}))$ in Fig 2) are defined to be $2\mathbf{x}_k$ and $(\pi - \mathbf{x}_j)(\pi - \mathbf{x}_k)$ respectively.

In other words, the input features are mapped non-linearly to a higher-dimensional Hilbert space using a feature mapping function. This function includes an exponential operator composed of non-linear functions and rotation Z operators. The initial two input samples are encoded into quantum states, and the resulting states are measured to determine whether the two samples share the same class label. This overlap is quantified by the fidelity function. The fidelity result is then used to update the neural network's weights, improving the separability of the two data classes into orthogonal subspaces and maximizing their difference.

Fig 5 shows the training loss plots (using *PyTorch* mean squared error package, *MSEloss*) as a function of the number of iterations for 9–12 principal components (PCs), during the training phase of the NQE method using the stochastic gradient descent algorithm to update the weights. This process resulted in an updated weight for the function $g(\boldsymbol{x}, \boldsymbol{w})$.

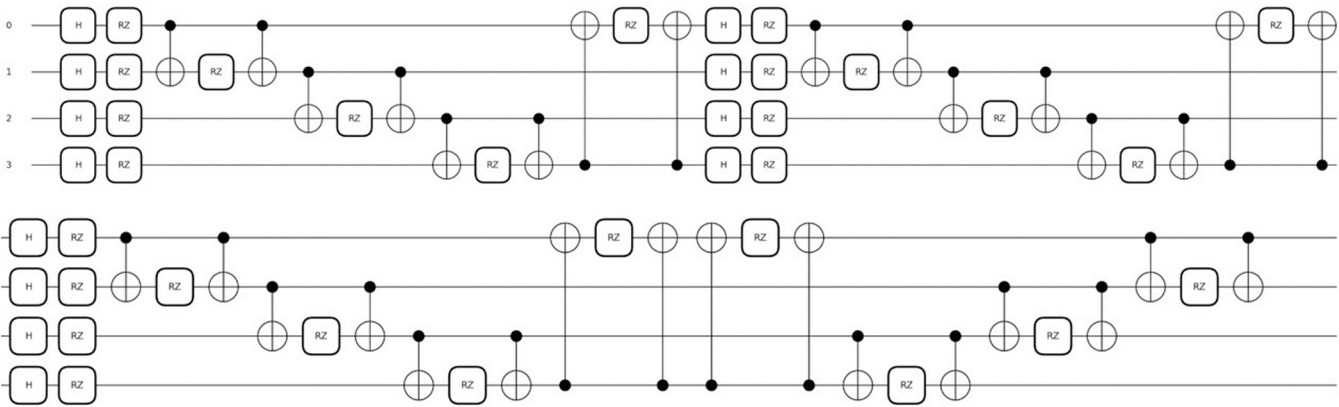

**Fig 4. The quantum embedding circuit designed for the miRNA dataset, featuring three layers of repeated Hadamard (H), RZ, and control-NOT unitary gates.** To conserve space, a partial depiction of the inverse RZ gates applied in reverse order is shown.

Once the weight for the function g($\boldsymbol{x}$, $\boldsymbol{w}$) is updated, it is used in operator V [Eq (1)], which changes the quantum circuit's parameters to better separate the two classes by maximizing their distance [Eq (2)]. A high training loss means the two classes are not well separated in the quantum space, so further adjustments are needed.

The training loss function plotting shows high training loss at the beginning which decreases during the training process and then flattens gradually. A high initial training loss indicates that the initial parameters of the NN are not good for modelling the data. As training progresses, the NN parameters are updated to minimize this loss, improving the model's performance. As the training loss flattens, it indicates that the NN is approaching convergence, meaning that further training may not result in significant improvements.

After achieving minimal training loss, one can generate a heatmap of the kernel matrix to visualize similarities between pairs of feature vectors mapped into a higher-dimensional space. Instead of presenting this intermediate result, our focus is on reporting the classification performance using the gene expression profile study. Visualization of the heatmap are deferred for the miRNA case study.

We utilize both SVC and QSVC on a quantum simulator within the IBM cloud through its quantum machine learning library, *Qiskit*, to assess and compare their respective performances.

The performance metrics comparison between NQE + SVC and NQE+QSVC for 9, 10, 11, and 12 PCs is illustrated in Fig 6. The first and second bar plots indicate the NQE + SVC and NQE+QSVC result, respectively. For recall (TPR), specificity (SPC), precision (PPV), F1 score, accuracy (ACC), negative predictive value (NPV), and Matthews Correlation Coefficient (MCC), higher scores indicate better performance. Conversely, for False Positive Rate (FPR), False Discovery Rate (FDR), and False Negative Rate (FNR), lower scores suggest better performance. As we can observe from Fig 6, NQE+QSVC performs better for 9 and 10 PCs for almost all the performance metric scores, on the other hand, classical SVC achieves better performance for 11 and 12 PCs.

We assessed the classification performance of SVC and QSVC models utilizing gene expression datasets from ccRCC cases. Table 1 presents a comparison of performance metric score of NQE + SVC and NQE+QSVC for each PC. Numbers highlighted in bold indicate the better performance. As shown in Table 1, NQE+QSVC outperforms NQE + SVC in two cases; i.e., both 9 and 10 PCs. When considering 9 and 10 PCs, NQE+QSVC achieved scores of 10 and 8 points, respectively, across the 10 performance metrics. Alternatively, NQE + SVC achieves better performance in two cases: 11 and 12 PCs. When evaluating 11 and 12 PCs, SVC attained scores of 8 and 6 points, respectively. Notably, NQE+QSVC achieves the highest F1-score and MCC of 0.447 and 0.304, respectively, for the 10 PCs case.

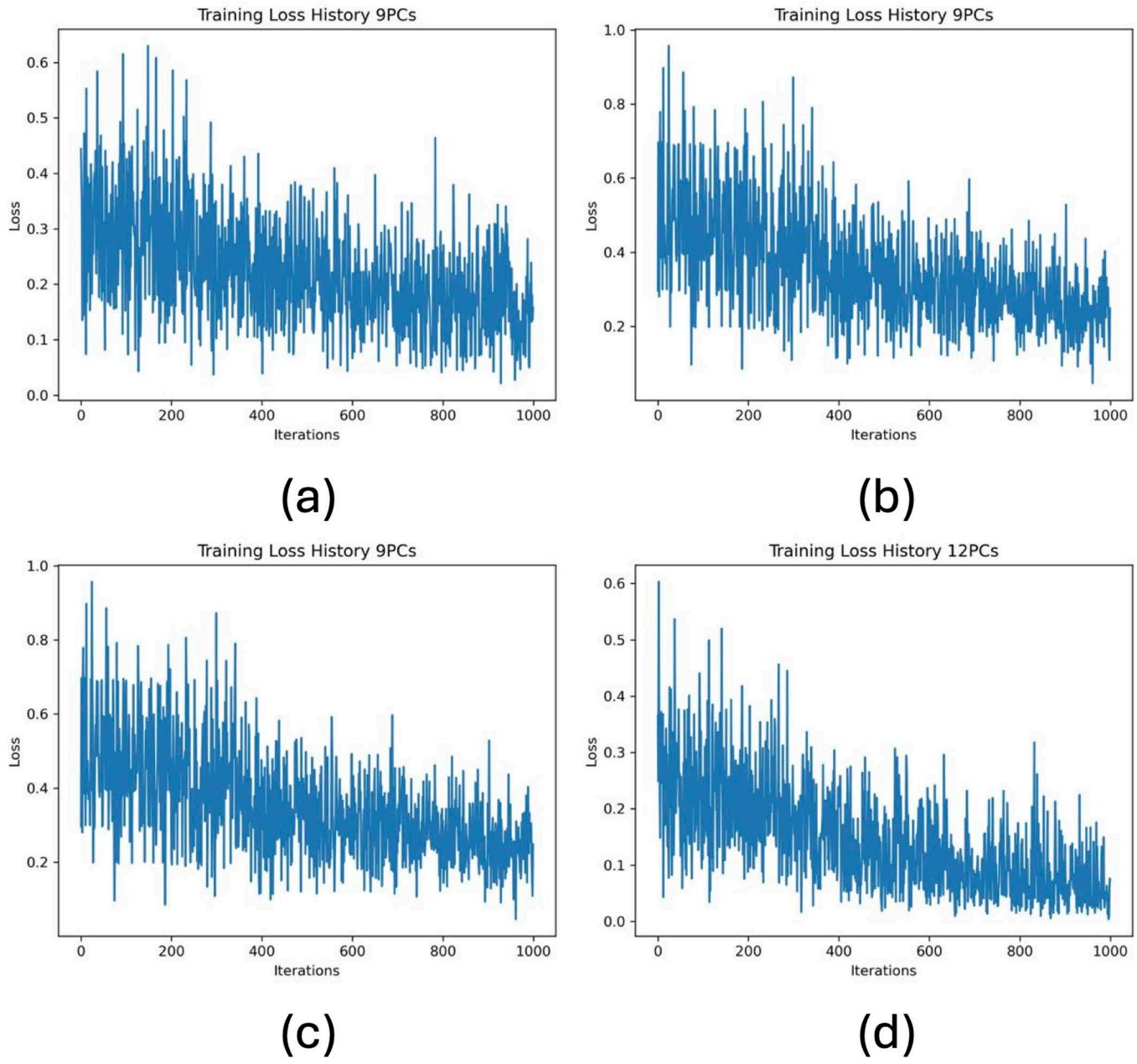

**Fig 5. Training loss results plotted against the number of iterations for (a) 9, (b) 10, (c) 11, and (d) 12 principal components.**

Next, we presented the results concerning the training loss and classification performance, employing the dataset comprising microRNA (miRNA) expression profiles of ccRCC patients.

We assess the classification performance of both SVC and QSVC with and without the utilization of NQE for data embedding. Without NQE, QSVC outperforms SVC. Specifically, in the comparison between SVC and QSVC, QSVC achieves scores of 8 points, one tier, across the 10 performance metrics (S3 File).

In addition, we have conducted a more comprehensive study by selecting the number of principal components (PCs) in the range of 4–6, for the classification analysis. Table 2 presents the results of the 10 performance metrics for the training

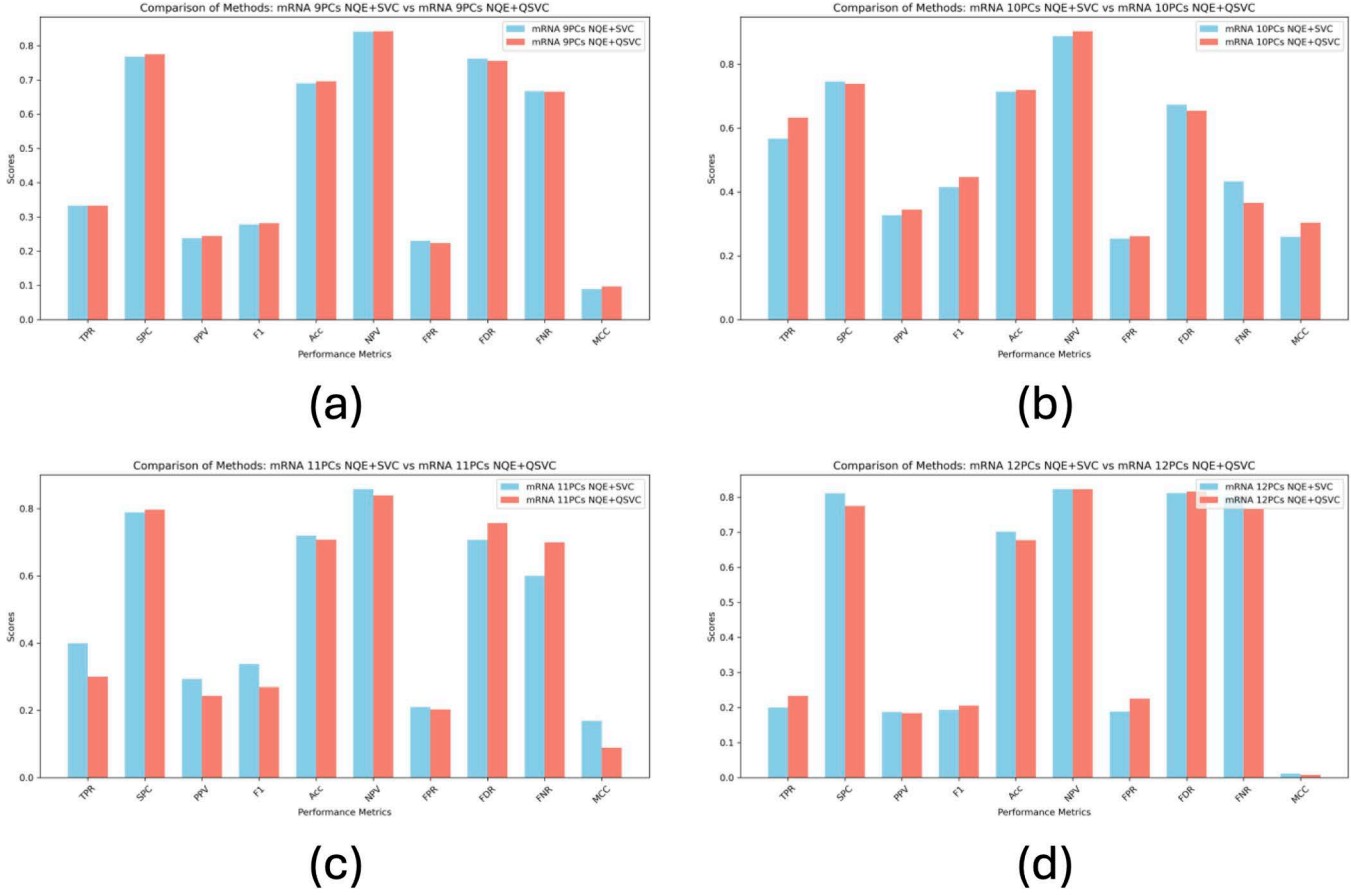

**Fig 6. The results of the ten performance metrics for NQE+SVC vs NQE+QSVC using (a) 9, (b) 10, (c) 11, and (d) 12 principle components, respectively.**

set, evaluated using five-fold cross-validation on the balanced training set, along with their standard deviations, providing a comprehensive view of the model's robustness.

The last row of Table 2 presents the comparative scores obtained by evaluating the 10 metrics, where higher values indicate better performance. We observed that NQE+QSVC outperforms NQE+SVC when using 5 and 6 PCs 'test' set results (comparative score is 0 vs. 10). However, NQE+SVC achieves superior performance with 4 PCs. These results suggest that NQE has the potential to improve the performance of QSVC classifiers. Additionally, all standard deviation values are relatively small in comparison to the mean values, except in five cases (NQE+SVC, 4PCs – SPC, FPR, FNR, and MCC; and NQE+SVC, 5PCs – SPC), indicating that the results for the 10 performance metrics are generally robust.

We also observed a significant difference between the performance metric values for the training and test sets, particularly for PPV, F1, FDR, FNR, and MCC. However, this difference is less pronounced when data balancing techniques are applied, compared to the scenario without data balancing, where the discrepancy between training and test set results is much more pronounced (Table 2). This suggests that the issue of data imbalance remains an inherently challenging problem.

For the miRNA case study using NQE, Fig 7 illustrates the loss function plot as a function of the number of iterations. The training loss function plot depicts initially high loss that diminishes over training, plateauing eventually. High initial loss suggests inadequate NN parameters for data embedding, gradually refined through training to enhance model

**Table 1. Comparison of performance metrics, SVC vs QSVC after using NQE for data embedding. Bold-faced fonts denote that the classifier achieves better performance.**

| Metrics | 9 PCs | | 10 PCs | | 11 PCs | | 12 PCs | |
|---|---|---|---|---|---|---|---|---|
| | SVC | QSVC | SVC | QSVC | SVC | QSVC | SVC | QSVC |
| Sensitivity/Recall, TPR | 0.333 | **0.333** | 0.567 | **0.633** | **0.4** | 0.3 | 0.2 | **0.233** |
| Specificity, SPC | 0.768 | **0.775** | **0.746** | 0.739 | 0.789 | **0.797** | **0.811** | 0.775 |
| Precision, PPV | 0.238 | **0.244** | 0.327 | **0.345** | **0.293** | 0.243 | **0.187** | 0.184 |
| F1 Score | 0.278 | **0.282** | 0.415 | **0.447** | **0.338** | 0.269 | 0.193 | **0.206** |
| Accuracy | 0.69 | **0.696** | 0.714 | **0.720** | **0.720** | 0.708 | **0.702** | 0.678 |
| Negative Predictive Value | 0.841 | **0.842** | 0.888 | **0.903** | **0.858** | 0.839 | **0.823** | 0.823 |
| False Positive Rate (FPR)* | 0.23 | **0.224** | **0.254** | 0.261 | 0.210 | **0.203** | 0.188 | **0.225** |
| False Discovery Rate (FDR)* | 0.762 | **0.756** | 0.673 | **0.654** | **0.707** | 0.757 | **0.812** | 0.816 |
| False Negative Rate (FNR)* | 0.667 | **0.666** | 0.433 | **0.366** | **0.6** | 0.7 | 0.8 | **0.766** |
| Matthews Correlation Coefficient | 0.089 | **0.097** | 0.259 | **0.304** | **0.169** | 0.089 | **0.011** | 0.008 |
| Comparative scores | 0 | 9 + 1 tier | 2 | 8 | 8 | 2 | 6 + 1 tier | 3 + 1 tier |

* A lower value implies better performance

**Table 2. Classification performance results using 5-fold cross-validation for the training datasets after applying the data imbalance tool, SMOTE and NQE for data embedding. The values without the '±' signs represent the results for the 'test' set.**

| Metrics | 4 PCs | | 5 PCs | | 6 PCs | |
|---|---|---|---|---|---|---|
| | SVC | QSVC | SVC | QSVC | SVC | QSVC |
| Sensitivity/Recall, TPR | 0.674±0.240, 0.5 | 0.707±0.058, 0.46 | 0.856±0.144, 0.36 | 0.81±0.033, 0.46 | 0.839±0.21, 0.4 | 0.819±0.061, 0.5 |
| Specificity, SPC | 0.543±0.348, 0.67 | 0.828±0.064, 0.64 | 0.29±0.193, 0.70 | 0.359±0.094, 0.71 | 0.678±0.229, 0.66 | 0.765±0.054, 0.69 |
| Precision, PPV | 0.638±0.13, 0.25 | 0.807±0.054, 0.22 | 0.552±0.038, 0.21 | 0.561±0.034, 0.26 | 0.743±0.099, 0.20 | 0.779±0.042, 0.26 |
| F1 Score | 0.612±0.15, 0.33 | 0.752±0.044, 0.30 | 0.66±0.089, 0.27 | 0.585±0.043, 0.34 | 0.758±0.188, 0.27 | 0.797±0.042, 0.34 |
| Accuracy | 0.608±0.089, 0.64 | 0.767±0.042, 0.61 | 0.573±0.049, 0.64 | 0.662±0.024, 0.66 | 0.758±0.11, 0.61 | 0.79±0.041, 0.66 |
| Negative Predictive Value (NPV) | 0.649±0.123, 0.86 | 0.739±0.040, 0.85 | 0.747±0.155, 0.84 | 0.648±0.063, 0.86 | 0.84±0.099, 0.84 | 0.812±0.052, 0.86 |
| False Positive Rate (FPR)* | 0.457±0.348, 0.33 | 0.172±0.064, 0.35 | 0.709±0.155, 0.29 | 0.64±0.094, 0.28 | 0.322±0.23, 0.33 | 0.235±0.054, 0.30 |
| False Discovery Rate (FDR)* | 0.36±0.13, 0.75 | 0.193±0.054, 0.77 | 0.448±0.038, 0.78 | 0.439±0.034, 0.74 | 0.256±0.099, 0.79 | 0.22±0.042, 0.74 |
| False Negative Rate (FNR)* | 0.33±0.13, 0.5 | 0.293±0.058, 0.53 | 0.144±0.144, 0.63 | 0.189±0.033, 0.53 | 0.16±0.211, 0.6 | 0.18±0.061, 0.5 |
| Matthews Correlation Coefficient (MCC) | 0.278±0.159, 0.14 | 0.541±0.082, 0.09 | 0.201±0.077, 0.06 | 0.188±0.088, 0.15 | 0.597±0.112, 0.05 | 0.587±0.082, 0.16 |
| Comparative scores | 10 | 0 | 0 | 10 | 0 | 10 |

* A lower value implies better performance

performance. Flattening loss indicates convergence, implying further training may yield limited improvements. This trend aligns with what is depicted in Fig 7. The interpretation of this plot is essentially identical to that of Fig 5.

Having achieved a minimal training loss, we proceed to create a kernel matrix, which is instrumental in the classification of the dataset. To accomplish this, we utilize the '*FidelityQuantumKernel*' class. During its instantiation, we provide two key

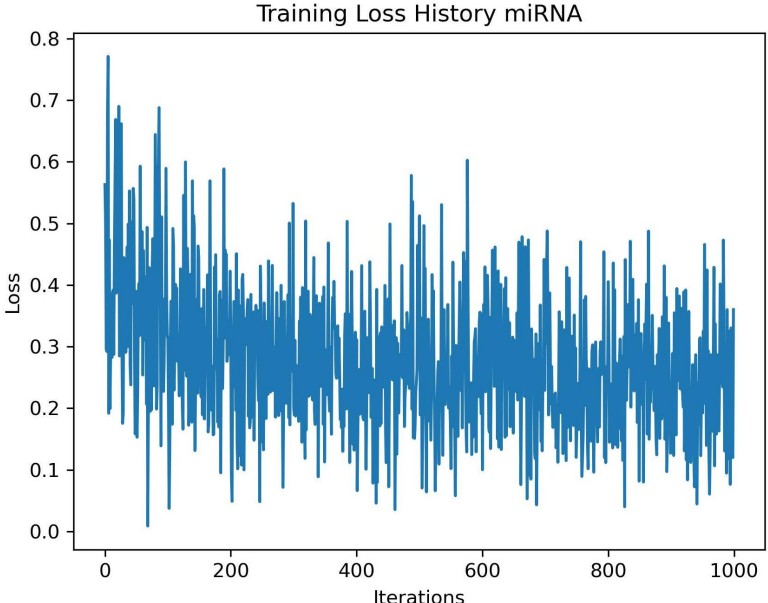

**Fig 7. Training loss results plotted against the number of iterations.**

parameters: first, the feature map, a two-qubit *ZZFeatureMap*, and second, the fidelity measure, employing the '*ComputeUncompute*' fidelity method that utilizes the '*Sampler*' primitive. The kernel matrix's elements reflect the similarity levels between pairs of feature vectors within a higher-dimensional space. This is achieved without the need to directly compute the mapping functions, thereby enabling us to identify a linear hyperplane that can be used for classification tasks within this transformed feature space. The train and test kernel matrices can be visualized as shown in Fig 8. Darker colored dots mean higher level of similarity of two feature vectors. The kernel matrix for the training set is a square, with both the x and y axes representing the size of the training set. For the testing set kernel matrix, the x-axis shows the training set size, and the y-axis shows the test set size.

As we can see from Fig 9, by using miRNA expression, NQE+QSVC performs better in every performance metric. The F1-score and MCC for NQE+QSVC is 0.92 and 0.68, whereas NQE + SVC achieves a score of 0.9 and 0.62. Again, lower scores for FPR, FDR, and FNR indicate better performance. In summary, NQE+QSVC attained higher scores of 9 points across all 10 performance metrics (S1 File), which is slightly better comparing to the case study where NQE is not used, that is, QSVC vs NQE+QSVC.

Table 3 presents the results of the 10 performance metrics for the training set, including standard deviation, after selecting the optimal cross-entropy value via 5-fold cross-validation. The final row of Table 3 provides the comparative scores across the 10 metrics using the 'test' set results, where higher values correspond to better performance. We observed that NQE + NN outperforms NQE + QNN when using 4 and 5 PCs. However, NQE + QNN demonstrates superior performance with 6 PCs, as indicated by the comparative score of 0 versus 10. These findings suggest that NQE has the potential to enhance the performance of the QNN classifiers. Furthermore, all standard deviation values are relatively small compared to the mean values, except in two instances (NQE + NN, 6 PCs, FPR, and FNR), indicating that the results for the 10 performance metrics are robust.

We also found a significant difference between the performance metric values for the training and test sets, particularly for PPV, F1, FDR, FNR, and MCC. As suggested earlier, the issue of data imbalance remains an intrinsically challenging problem.

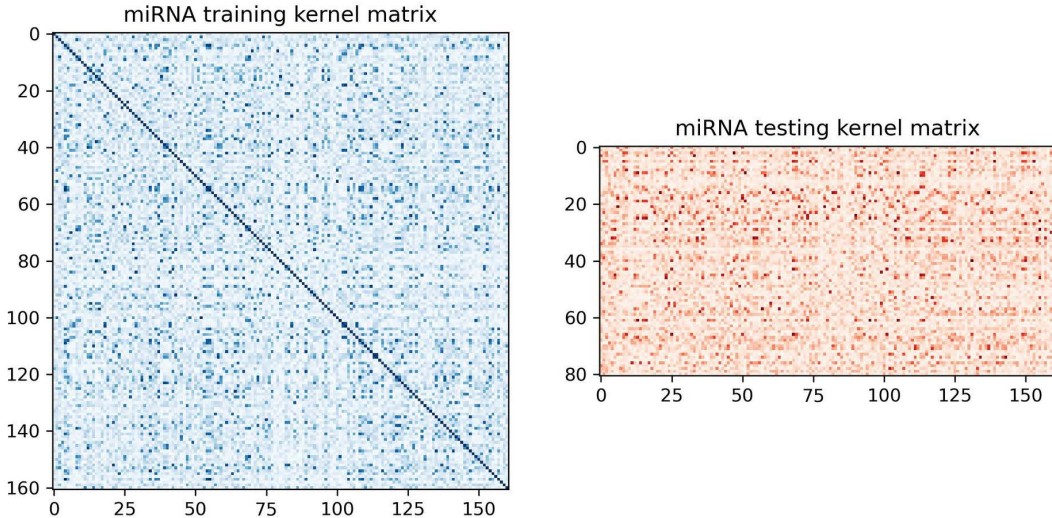

**Fig 8. The train and test kernel matrix matrices.** Darker colored dots mean higher level of similarity.

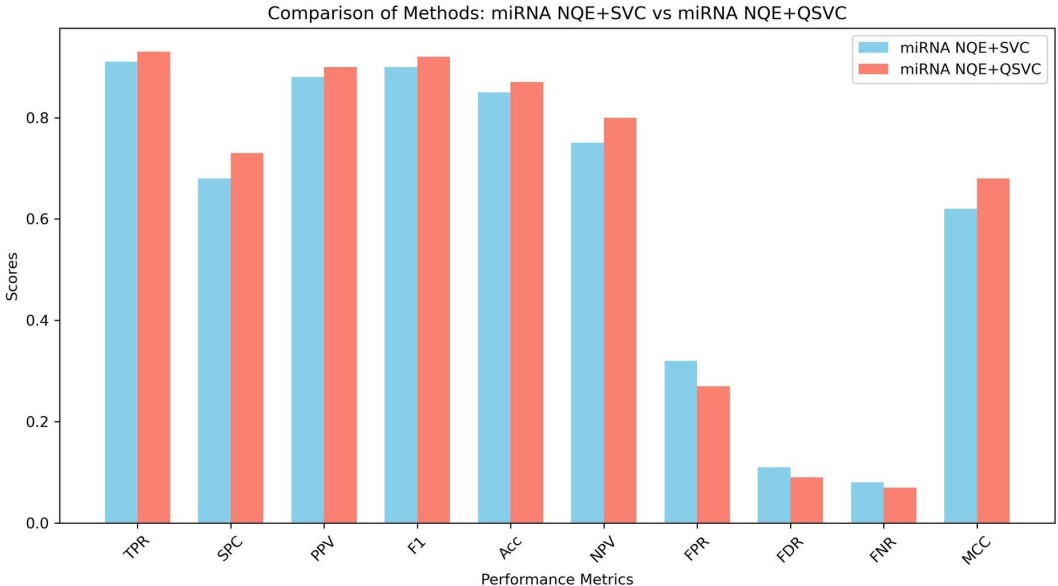

**Fig 9. The results of the ten performance metrics for NQE+SVC vs NQE+QSVC based on classifying miRNA expression profiles.**

To evaluate how well our method works, we applied it to a third dataset. Our results show that among the three case studies NQE+QSVC outperforms NQE + SVC in two cases, while NQE + NN performs better than NQE + QNN in two cases (Table 4). Since the analysis process is the same as with the first two datasets, we only highlight the main findings and provide the full results in the supplementary material.

S4 File as well as S5–S7 Figs show the results of 10 performance measurements for the training and test sets of dataset (iii), evaluated using NQE+SVC and NQE+QSVC. These results include standard deviation and are based on choosing the best cross-entropy value using 5-fold cross-validation. The last row of the table shows the test set results across the 10

**Table 3. Classification performance results using 5-fold cross-validation for the training datasets after applying the data imbalance tool, SMOTE and NQE for data embedding. The values without the '±' signs represent the results for the 'test' set.**

| Metrics | 4 PCs | | 5 PCs | | 6 PCs | |
|---|---|---|---|---|---|---|
| | NN | QNN | NN | QNN | NN | QNN |
| Sensitivity/Recall, TPR | 0.727 ±0.064 0.433 | 0.676 ±0.056 0.433 | 0.812 ±0.045 0.567 | 0.73 ±0.08 0.467 | 0.772 ±0.104 0.533 | 0.641 ±0.09 0.633 |
| Specificity, SPC=TNR | 0.799 ±0.078 0.725 | 0.766 ±0.07 0.681 | 0.733 ±0.086 0.623 | 0.706 ±0.082 0.667 | 0.687 ±0.103 0.449 | 0.632 ±0.077 0.645 |
| Precision, PPV | 0.79 ±0.057 0.255 | 0.748 ±0.052 0.228 | 0.758 ±0.055 0.246 | 0.717 ±0.047 0.233 | 0.717 ±0.051 0.174 | 0.637 ±0.042 0.279 |
| F1 Score | 0.754 ±0.033 0.321 | 0.708 ±0.035 0.299 | 0.782 ±0.029 0.343 | 0.72 ±0.046 0.311 | 0.738 ±0.054 0.262 | 0.636 ±0.053 0.388 |
| Accuracy | 0.763 ±0.031 0.673 | 0.721 ±0.033 0.637 | 0.772 ±0.036 0.613 | 0.718 ±0.039 0.631 | 0.73 ±0.047 0.464 | 0.637 ±0.04 0.643 |
| Negative Predictive Value | 0.748 ±0.035 0.855 | 0.704 ±0.033 0.847 | 0.797 ±0.031 0.869 | 0.727 ±0.048 0.852 | 0.762 ±0.074 0.816 | 0.642 ±0.05 0.89 |
| False Positive Rate (FPR)* | 0.201 ±0.078 0.275 | 0.234 ±0.07 0.319 | 0.267 ±0.086 0.377 | 0.294 ±0.082 0.333 | 0.313 ±0.103 0.551 | 0.368 ±0.077 0.355 |
| False Discovery Rate (FDR)* | 0.21 ±0.057 0.745 | 0.252 ±0.052 0.772 | 0.242 ±0.055 0.754 | 0.283 ±0.047 0.767 | 0.283 ±0.051 0.826 | 0.363 ±0.042 0.721 |
| False Negative Rate (FNR)* | 0.273 ±0.064 0.567 | 0.324 ±0.056 0.567 | 0.188 ±0.045 0.433 | 0.27 ±0.08 0.533 | 0.228 ±0.104 0.467 | 0.359 ±0.09 0.367 |
| Matthews Correlation Coefficient | 0.532 ±0.062 0.132 | 0.447 ±0.068 0.093 | 0.549 ±0.068 0.148 | 0.44 ±0.078 0.107 | 0.468 ±0.092 −0.013 | 0.276 ±0.082 0.217 |
| Comparative scores | 8+2-tier | 2- tier | 7 | 3 | 0 | 10 |

*A lower value implies better performance

**Table 4. Comparison of NQE combined with SVC, NN, and QCNN classifiers. The table shows the number of test cases, comparison with classical classifiers, and data imbalance issues in this and previous work.**

| | SVC | QSVC | NN | QNN | QCNN |
|---|---|---|---|---|---|
| Tested dataset | Dataset (i), mRNA | Dataset (i), mRNA | Dataset (i), mRNA | Dataset (i), mRNA | MNIST |
| Tested cases | 7 | 7 | 3 | 3 | 3 |
| Better performance | 3 | 4 | 1 | 2 | Not available |
| Tested dataset | Dataset (ii), miRNA | Dataset (ii), miRNA | Dataset (ii), miRNA | Dataset (ii), miRNA | Fashion-MNIST |
| Tested cases | 1 | 1 | null | null | 3 |
| Better performance | 0 | 1 | | | Not available |
| Tested dataset | Dataset (iii), miRNA | Dataset (iii), miRNA | Dataset (iii), miRNA | Dataset (iii), miRNA | |
| Tested cases | 3 | 3 | 3 | 3 | |
| Better performance | 1 | 2 | 1+1 tier | 1+1 tier | |
| Compare with classical ML | yes | yes | yes | yes | no |
| Data imbalance issue | yes | yes | yes | yes | no |

metrics. Higher values mean better performance, except for FPR, FDR, and FNR, where lower values are better. We found that NQE+QSVC did better than NQE + SVC when using 5 and 6 PCs. For example, the scores were 0 vs. 10 and 3 vs. 7. This suggests that NQE can help improve the QSVC model. Most standard deviation values are small compared to their averages, which means the results are quite stable. Only FNR and MCC had larger variation. We also noticed a big difference between training and test scores, especially for MCC. This shows that the issue of unbalanced data is still a hard problem to solve.

S5 File and S8–S10 Figs present the results of 10 performance metrics for the training and test sets of dataset (iii), analysed using NQE + NN and NQE + QNN. We found that NQE + QNN did better than NQE + SVC when using 6 PCs, and both NQE + NN and NQE + QNN perform equally well when using 4 PCs.

## 4. Discussion

The application of NQE to enhance machine learning classification was initially explored with non-biomedical data, specifically the MNIST dataset. The results indicate that combining NQE with QCNN improves classification performance. However, there is no comparison with NQE + CNN to determine if it yields inferior results compared to the NQE+QCNN architecture.

We explored the application of NQE to enhance machine learning classification in four ways: (i) using three types of biomedical data, (ii) addressing data imbalance issues, (iii) comparing NQE+QSVC with NQE + SVC, and (iv) comparing NQE + QNN with NQE + NN. This allows us to determine whether machine learning classification algorithms yield inferior results without the use of NQE compared to when NQE is utilized. Our results indicate that the NQE approach can be extended to molecular medicine data, demonstrating its potential as a general method to enhance classification performance. This study does not aim to improve the data embedding process or the quantum machine learning algorithms.

We evaluate the performance of SVC/NN compared to QSVC/QNN after applying NQE for classifying patients with ccRCC. This classification is based on their normal, metastasis and primary status using three independent datasets.

From our results, the following four findings were established. First, by utilizing the PCA method, the required qubits decreases; hence, allow us to complete the classification task within reasonable time. This process effectively dealing with high-dimensional datasets enhancing the feature extraction process. Second, upon performance comparison, NQE+QSVC (NQE + QNN) demonstrates better prediction capabilities compared to NQE + SVC (NQE + NN) in certain cases. Without the use of NQE, QSVC/QNN performs relatively poorly compared to SVC/NN. We demonstrate the effectiveness of the NQE approach across three biomedical datasets. Third, these results have potential clinical implication in metastasis biomarker study for precision medicine diagnosis. It can aid physicians in making accurate diagnoses and minimizing the wastage of medical resources. Fourth, we demonstrate the effectiveness and generalizability of the current approach by analysing three independent biomedical datasets. This suggests that the approach could be applicable to other datasets as well.

Some might be concerned about comparing our hybrid approach to other machine learning classifiers to evaluate NQE's effectiveness. As mentioned earlier, only the quantum versions of SVC, NN, and CNN are available in IBM Qiskit. Table 4 presents a comparison of NQE combined with these three classifiers, highlighting the strengths and weaknesses of each method. We tested QSVC thoroughly using three datasets and seven different test cases (using different PCs). This means our method was checked more carefully than the others.

We also compared how well two QML models worked against classical machine learning models. The results showed that combining NQE with the *quantum advantage* of QML is an effective approach. On the other hand, the work of QCNN has not been compared to CNN to see if it performs better [41]. Furthermore, we address the issue of data imbalance, which is not considered in the previous work as well. Out of the 17 test cases, NQE+QSVC and NQE + QNN produce better results in 10 cases and are tied in one case compared to NQE + SVC and NQE + NN. We note that out of the 10 winning cases, NQE+QSVC accounts for seven, suggesting that it performs better overall.

Over several decades, clear cell renal cell carcinoma (ccRCC) has been the predominant subtype of renal cancer, accounting for 80–90% of renal cases and remaining a highly lethal urological malignancy. Metastasis, the final stage of cancer, remains a challenging area of study, with the precise mechanisms behind cancer cell dissemination and

establishment at distant sites still largely unknown. In this context, both gene and microRNA (miRNA) expression profiles are used to classify tumor metastasis, offering a more accurate and comprehensive patient classification. This approach aligns with the principles of precision medicine, tailoring treatments based on molecular profiles to improve outcomes. By combining both mRNA and miRNA data, this study helps us understand cancer metastasis better. It also supports creating more personalized treatments for ccRCC and other types of cancer.

There is a limitation of utilizing the gene expression dataset for patients with ccRCC from the UCSC Xena database. The most obvious limitation is that there is data imbalance problem, where the M0 samples being 5.44 times larger than the M1 samples. In case of using the miRNA expression profiles, the overall performance for the 10 metrics is better than the use of the gene expression data. For example, using miRNA expression data, the NQE+QSVC model achieved an F1-score of 0.92 and an MCC of 0.68. In comparison, with gene expression data, the scores were lower—0.447 for F1 and 0.304 for MCC. This variance could be attributed to the utilization of the data imbalance package, SMOTE, which likely has a significant influence on classification performance. In fact, the imbalance between the number of M0 and M1 cases is a prevalent issue observed in at least 14 cancer types documented by the Xena database. This limitation affects classification performance even if one chooses a different cancer cohort.

Another limitation is that simulating quantum circuits on classical computers is constrained by scalability issues. Even when encoding features with only a few quantum qubits, the process remains highly time-consuming. A promising solution involves using tensor networks to represent quantum circuits. To address this, we plan to accelerate the QSVC implementation with NVIDIA's cuQuantum package [77]. We aim to enhance computational efficiency by integrating Tensor Network (TN) technique into the QSVC implementation. TN technique enables efficient simulation by converting quantum circuits into tensor-network representations and optimizing the contraction sequence to reduce both time complexity and memory usage. This future work could demonstrate the potential for scaling to more than 30 qubits using tensor-based optimization, enabling efficient and accurate analysis of high-dimensional biomedical data.

## 5. Conclusion

We propose a classical-quantum approach for the binary classification problem. To ensure a fair comparison between the SVC/NN and QSVC/QNN classification results, we perform hyperparameter optimization to eliminate any bias towards a particular classification method.

Additionally, we aim to assess whether combining NQE with QSVC works better than combining NQE with SVC or NN on three biomedical datasets. Our results show that the proposed method is effective. To summarize, NQE+QSVC/NQE+QNN exhibits better performance than NQE+SVC/NQE+NN across a range of performance metrices. These results could have potential clinical implication in metastasis biomarker study for precision medicine diagnosis.

## Supporting information

**S1 File. The results of the 48 differentially expressed genes obtained by analysing gene expression profiles using the DEseq2 package.**
(CSV)

**S2 File. The results of the 4 differentially expressed miRNAs obtained by analysing miRNAs expression profiles using the R 'limma' package.**
(CSV)

**S3 File. The results of the 10 performance metrics of the miRNA biomarker study (NT vs M0) for the four case studies are as follows: SVC, QSVC, NQE+SVC, and NQE+QSVC.**
(DOCX)

**S4 File. The results of the 10 performance metrics of the miRNA biomarker study (M0 vs M1) for the two case studies are as follows: NQE+SVC, and NQE+QSVC.**
(DOCX)

**S5 File. The results of the 10 performance metrics of the miRNA biomarker study (M0 vs M1) for the two case studies are as follows: NQE+NN, and NQE+QNN.**
(DOCX)

**S1 Fig. The PCA plots of the data distribution (mRNA) projected onto two principal components selected from PC1 through PC9.**
(DOCX)

**S2 Fig. The PCA plots of the data distribution (mRNA) projected onto two principal components selected from PC1 through PC10.**
(JPEG)

**S3 Fig. The PCA plots of the data distribution (mRNA) projected onto two principal components selected from PC1 through PC11.**
(JPEG)

**S4 Fig. The PCA plots of the data distribution (mRNA) projected onto two principal components selected from PC1 through PC12.**
(JPEG)

**S5 Fig. The results of 10 performance metrics from the miRNA biomarker study (M0 vs M1), comparing NQE+SVC and NQE+QSVC in the 4 PCs case.**
(ZIP)

**S6 Fig. The results of 10 performance metrics from the miRNA biomarker study (M0 vs M1), comparing NQE+SVC and NQE+QSVC in the 5 PCs case.**
(PNG)

**S7 Fig. The results of 10 performance metrics from the miRNA biomarker study (M0 vs M1), comparing NQE+SVC and NQE+QSVC in the 6 PCs case.**
(PNG)

**S8 Fig. The results of 10 performance metrics from the miRNA biomarker study (M0 vs M1), comparing NQE+NN and NQE+QNN in the 4 PCs case.**
(PNG)

**S9 Fig. The results of 10 performance metrics from the miRNA biomarker study (M0 vs M1), comparing NQE+NN and NQE+QNN in the 5 PCs case.**
(PNG)

**S10 Fig. The results of 10 performance metrics from the miRNA biomarker study (M0 vs M1), comparing NQE+NN and NQE+QNN in the 6 PCs case.**
(PNG)

## Author contributions

**Conceptualization:** Pin-Keng Shih, Ng Ka-Lok.

**Data curation:** Venugopala Reddy Mekala, Ezra B. Wijaya.

**Formal analysis:** Aninda Astuti, Venugopala Reddy Mekala, Ezra B. Wijaya.

**Funding acquisition:** Pin-Keng Shih.

**Investigation:** Aninda Astuti, Shan-Chih Lee, Ezra B. Wijaya.

**Methodology:** Aninda Astuti, Ng ka-Lok.

**Project administration:** Pin-Keng Shih, Ng Ka-Lok.

**Validation:** Aninda Astuti, Shan-Chih Lee, Venugopala Reddy Mekala.

**Visualization:** Aninda Astuti.

**Writing – original draft:** Ng Ka-Lok.

**Writing – review & editing:** Ng Ka-Lok.

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
