## [Decision Letter · Decision Letter 0]

Dear Dr. KALOK,

Thank you for submitting your manuscript to PLOS ONE. After careful consideration, we feel that it has merit but does not fully meet PLOS ONE’s publication criteria as it currently stands. Therefore, we invite you to submit a revised version of the manuscript that addresses the points raised during the review process.

We look forward to receiving your revised manuscript.

Kind regards,

Richard Jiang

Academic Editor

PLOS ONE

“Miss Aninda Astuti and Dr. Ka-Lok Ng works are supported by (i) National Science and Technology Council (NSTC), Taiwan (grant numbers: NSTC 112-2221-E-468-021), and (ii) Asia University and China Medical University Hospital (grant number: ASIA-112-CMUH-6). The funding bodies were not involved in the design of the study, in the collection, analysis of the data, or in writing of the manuscript.

During the preparation of this work, the author, Ka-Lok Ng, used ChatGPT and Bing for grammar correction, and content revision of our English writing. After using this tool/service, the author, Ka-Lok Ng, reviewed and edited the content as needed and take full responsibility for the content of the publication. We utilized the 'Turnitin' package to detect plagiarism in our writing, revealing a 16% similarity level.”

“Miss Aninda Astuti and Dr. Ka-Lok Ng works are supported by (i) National Science and Technology Council (NSTC) (grant numbers: NSTC 112-2221-E-468-021), and (ii) Asia University and China Medical University Hospital (grant number: ASIA-112-CMUH-6). The funding bodies were not involved in the design of the study, in the collection, analysis of the data, or in writing of the manuscript.”

3. Please remove your figures from within your manuscript file, leaving only the individual TIFF/EPS image files, uploaded separately. These will be automatically included in the reviewers’ PDF.

Reviewers' comments:

Reviewer's Responses to Questions

**Comments to the Author**

1. Is the manuscript technically sound, and do the data support the conclusions?

Reviewer #1: Partly

Reviewer #2: Partly

2. Has the statistical analysis been performed appropriately and rigorously?

Reviewer #1: No

Reviewer #2: Yes

3. Have the authors made all data underlying the findings in their manuscript fully available?

Reviewer #1: Yes

Reviewer #2: Yes

4. Is the manuscript presented in an intelligible fashion and written in standard English?

Reviewer #1: Yes

Reviewer #2: No

Reviewer #1: This paper presents an innovative application of quantum machine learning (QML) in cancer biomarker classification. The paper explores the use of a hybrid quantum-classical approach to improve biomarker classification in metastatic cancer, specifically clear cell renal cell carcinoma (ccRCC). The authors aim to enhance the performance (accuracy and execution time) of the Quantum Support Vector Classifier (QSVC) by integrating it with Neural Quantum Embedding (NQE) and compare its performance with a classical Support Vector Classifier (SVC).

This paper focuses on evaluating the effectiveness of QSVC, combined with NQE, for binary classification of cancer data.

The integration of NQE and QSVC is an innovative approach, particularly for biomedical datasets like gene expression and miRNA profiles related to cancer metastasis. This combination of quantum and classical techniques shows promise for complex classification tasks but broader dataset testing and better explanations of quantum principles could enhance its impact.

While the paper presents a novel and promising approach, but in order to strengthen the paper and address potential concerns from readers, I recommend a major revision aim to clarify certain aspects of the methodology, improve transparency, and provide a deeper understanding of presented concepts:

1. Expand on Quantum Concepts: While the research is highly technical and relevant to a specialized audience, it may be too complex for a broader readership. There could be more effort in explaining the underlying quantum principles for readers less familiar with quantum computing, which would make the paper more accessible.

2. Could you provide more detailed explanations or visual aids to clarify how the quantum embedding circuit (Figure 3) functions for readers unfamiliar with quantum mechanics? A simplified diagram or intuitive description might enhance accessibility.

3. Could you explain more intuitively how the quantum embedding circuit contributes to the improvement in classification performance? For readers unfamiliar with quantum computing, an intuitive description would be helpful.

4. Why did you specifically choose the QSVC and NQE hybrid approach for your study? While the study compares QSVC and SVC, it would benefit from comparing additional classical machine learning methods (e.g., random forest, neural networks) to better position the advantages of the hybrid approach.

5. The manuscript provides mean performance metrics, but what is the variability (e.g., standard deviation, confidence intervals) across different cross-validation folds? Reporting this would offer a more complete picture of model robustness.

6. The manuscript notes that the M0 samples (non-metastasis) are 5.44 times larger than M1 samples (metastasis). Have you considered performing analyses without data balancing to compare results with and without SMOTE to assess the real impact of data imbalance? What measures did you take to ensure that SMOTE did not introduce bias?

7. Why did you select principal components ranging from 9 to 12 for the gene expression study? Did you experiment with other numbers of principal components, and how did those results compare? Could you explain more about the rationale behind this choice?

8. The study focuses on two specific datasets (gene expression and miRNA profiles). Have you tested the NQE+QSVC model on other biomedical datasets or non-biomedical data to assess its generalizability?

9. How does the use of SMOTE (Synthetic Minority Oversampling Technique) affect the results, especially in terms of potential overfitting? Have you considered alternative data balancing techniques (e.g., undersampling or cost-sensitive learning), and how would they compare to SMOTE in your study?

For Future Work:

1. Do you have plans to implement the QSVC model on real quantum hardware (e.g., IBM Q or other NISQ devices)? If so, how do you plan to address potential challenges like noise and qubit limitations?

2. Given that your study is focused on improving metastasis classification, how do you envision your results being used in clinical settings? Are there particular areas in precision medicine where this hybrid quantum approach could have an immediate impact?

Reviewer #2: The research explores the application of QML in identifying gene expression biomarkers for metastasis in ccRCC, a highly lethal form of renal cancer. The study focuses on improving execution time and accuracy in binary classification using the Quantum Support Vector Classifier compared to the classical SVC. Inspired by the Neural Quantum Embedding (NQE) method, they propose a two-stage approach to evaluate whether combining NQE with QSVC outperforms NQE with SVC across various biomedical datasets.

1. The organization of the paper needs to be improved, especially the Methods and Results sections.

2. Although the article uses quantum computing and an interesting dataset, the research method lacks novelty in both the data embedding part and the quantum support vector machine. I am more looking forward to seeing improvements and enhancements to these methods.

3. Figure 1 and figure 2 need to be improved, they are not very clear at the moment.

4. There is some lack of mathematical description of quantum support vector machines.

5. The loss function in figure 4 is quite confusing and I don’t know what it reflects.

**Do you want your identity to be public for this peer review?** For information about this choice, including consent withdrawal, please see our Privacy Policy

Reviewer #1: No

Reviewer #2: No

---

## [Author Response · Author response to Decision Letter 1]

8 Jan 2025

Dear Reviewers,

We thank the reviewers for their generous comments on the manuscripts and have revised the manuscript to address their concerns.

A marked-up copy and a clean copy have uploaded. Also, a revised version of ‘Figures (list of figures)’ and Supplementary files have uploaded.

Yours Sincerely,

Ka-Lok NG

Distinguish Professor

Dept. of Bioinformatics and Medical Engineering

Asia University

Taiwan

---

## [Decision Letter · Decision Letter 1]

Dear Dr. KALOK,

Thank you for submitting your manuscript to PLOS ONE. After careful consideration, we feel that it has merit but does not fully meet PLOS ONE’s publication criteria as it currently stands. Therefore, we invite you to submit a revised version of the manuscript that addresses the points raised during the review process.

We look forward to receiving your revised manuscript.

Kind regards,

Richard Jiang

Academic Editor

PLOS ONE

Additional Editor Comments:

Please follow the review comments and make the revision carefully.

Reviewers' comments:

Reviewer's Responses to Questions

**Comments to the Author**

Reviewer #1: All comments have been addressed

Reviewer #3: (No Response)

2. Is the manuscript technically sound, and do the data support the conclusions?

Reviewer #1: Yes

Reviewer #3: Partly

3. Has the statistical analysis been performed appropriately and rigorously?

Reviewer #1: (No Response)

Reviewer #3: N/A

4. Have the authors made all data underlying the findings in their manuscript fully available?

Reviewer #1: Yes

Reviewer #3: No

5. Is the manuscript presented in an intelligible fashion and written in standard English?

Reviewer #1: Yes

Reviewer #3: Yes

Reviewer #1: Dear Authors,

Thank you for addressing my previous comments and significantly improving the manuscript. However, I have a few minor suggestions to further enhance the clarity and reproducibility of your work.

- Please provide a direct link about the dataset used in this study to facilitate reproducibility.

- Consider sharing the implementation code (e.g., via GitHub or a public repository) to support further validation and adoption of your approach.

Reviewer #3: 1. The revised manuscript has addressed some reviewer comments; however, significant issues remain. First, the explanation of the training loss (as presented in Figure 4) is still too terse. The manuscript merely states that the loss decreases over iterations and “results in an updated weight for the function ( , )” without providing an intuitive explanation of what this loss represents regarding convergence or overall model performance. Please expand this discussion, clarifying how the loss behavior reflects improvements in training and its implications for the robustness of the quantum embedding.

2. Although the authors have added details regarding the quantum embedding circuit, the description remains highly verbose. It is recommended that the authors provide a more accessible, top-level overview of the embedding process. This should include a schematic or algorithmic flow diagram that distinguishes clearly between the novel contributions (e.g., the integration of NQE with QSVC/QNN) and the components based on existing methods, along with a discussion of the design decisions made.

3. The manuscript still suffers from language and organizational issues. Several sections are laden with technical jargon and complex phrasing, which hinders clarity. A thorough language and structural revision is necessary to improve readability and ensure consistency throughout the text.

4. The presentation of the experimental results, especially in Tables 1–3 and Figures 4–8, would benefit from a more detailed discussion. Each figure and table should be accompanied by a list of key observation points that help the reader interpret the significance of the performance metrics. This would also aid in understanding the practical impact of the proposed method.

5. The description of the experimental setup and workflow—including the use of data preprocessing techniques (such as SMOTE and PCA), hyperparameter tuning, and the simulation of quantum circuits—remains insufficient. More details should be provided to allow reproducibility. It is strongly encouraged that the authors share their source code via an online repository.

6. While the manuscript compares the performance of NQE+QSVC versus NQE+SVC (and similarly NQE+QNN versus NQE+NN), it would be valuable to include a discussion of how these hybrid approaches compare with more conventional classical methods beyond the SVC and NN models presented. Please discuss the limitations of the related work and explain in detail how the proposed framework overcomes these issues.

7. The current experimental evaluation is limited to two datasets (gene expression and miRNA expression profiles for ccRCC metastasis). To demonstrate the generalizability of the proposed method, the authors should consider applying their framework to a broader range of more challenging benchmarks.

8. Finally, please discuss the potential impact of this work at a larger scale and outline possible future research directions. Addressing how this hybrid quantum-classical approach could be applied to other domains or scaled up for more complex applications would significantly strengthen the manuscript.

**Do you want your identity to be public for this peer review?** For information about this choice, including consent withdrawal, please see our Privacy Policy

Reviewer #1: No

Reviewer #3: No

---

## [Author Response · Author response to Decision Letter 2]

4 Jun 2025

We thank the reviewers for their generous comments on the manuscripts and have revised the manuscript to address their concerns. Please refer to our 'Response to Reviewers' letter and the highlighted text in the 'Revised Manuscript with Tracked Changes' included in our submission.

---

## [Decision Letter · Decision Letter 2]

Use of Hybrid Quantum-Classical Algorithms for Enhancing Biomarker Classification

PONE-D-24-31594R2

Dear Dr. KALOK,

We’re pleased to inform you that your manuscript has been judged scientifically suitable for publication and will be formally accepted for publication once it meets all outstanding technical requirements.

Kind regards,

Richard Jiang

Academic Editor

PLOS ONE

Additional Editor Comments (optional):

Reviewers' comments:

Reviewer's Responses to Questions

**Comments to the Author**

Reviewer #3: All comments have been addressed

2. Is the manuscript technically sound, and do the data support the conclusions?

Reviewer #3: (No Response)

3. Has the statistical analysis been performed appropriately and rigorously?

Reviewer #3: (No Response)

4. Have the authors made all data underlying the findings in their manuscript fully available?

Reviewer #3: (No Response)

5. Is the manuscript presented in an intelligible fashion and written in standard English?

Reviewer #3: (No Response)

Reviewer #3: The authors addressed the comments. The manuscript can be accepted.

**Do you want your identity to be public for this peer review?** For information about this choice, including consent withdrawal, please see our Privacy Policy

Reviewer #3: No

---

## [Editor Report · Acceptance letter]

PONE-D-24-31594R2

PLOS ONE

Dear Dr. KALOK,

I'm pleased to inform you that your manuscript has been deemed suitable for publication in PLOS ONE. Congratulations! Your manuscript is now being handed over to our production team.

Kind regards,

on behalf of

Dr. Richard Jiang

Academic Editor

PLOS ONE